# Reactive Power Management Considering Stochastic Optimization under the Portuguese Reactive Power Policy Applied to DER in Distribution Networks

**Tiago Abreu** [1], **Tiago Soares** [1,*], **Leonel Carvalho** [1], **Hugo Morais** [2], **Tiago Simão** [3] and **Miguel Louro** [3]

1    INESC TEC - INESC Technology and Science, Centre for Power and Energy Systems,
4200-465 Porto, Portugal; tiago.j.abreu@inesctec.pt (T.A.); leonel.m.carvalho@inesctec.pt (L.C.)
2    INESC-ID, Department of Electrical and Computer Engineering, Instituto Superior Técnico-IST,
Universidade de Lisboa, 1049-001 Lisbon, Portugal; hugo.morais@tecnico.ulisboa.pt
3    EDP Distribution; 1050-121 Lisbon, Portugal; TiagoFilipe.Simao@edp.pt (T.S.); Miguel.Louro@edp.pt (M.L.)
*    Correspondence: tiago.a.soares@inesctec.pt

**Abstract:** Challenges in the coordination between the transmission system operator (TSO) and the distribution system operator (DSO) have risen continuously with the integration of distributed energy resources (DER). These technologies have the possibility to provide reactive power support for system operators. Considering the Portuguese reactive power policy as an example of the regulatory framework, this paper proposes a methodology for proactive reactive power management of the DSO using the renewable energy sources (RES) considering forecast uncertainty available in the distribution system. The proposed method applies a stochastic sequential alternative current (AC)-optimal power flow (SOPF) that returns trustworthy solutions for the DSO and optimizes the use of reactive power between the DSO and DER. The method is validated using a 37-bus distribution network considering real data. Results proved that the method improves the reactive power management by taking advantage of the full capabilities of the DER and by reducing the injection of reactive power by the TSO in the distribution network and, therefore, reducing losses.

**Keywords:** decision-aid; distributed energy resources; distribution system operator; reactive power management; uncertainty

## 1. Introduction

The implementation of renewable energy sources (RES) and the deployment of distributed energy resources (DER) have created a trend of evolution in the distribution network that requires the adaptation of the conventional practices to handle the behavior that is related to the RES [1]. These new procedures are compelled to have a more proactive role by the distribution system operator (DSO), controlling and/or contracting DER to deal with voltage and line/transformers congestions problems [2]. Incorporating forecast in the system operation of the DSO, as well as creating contract services with the DER, where the power flexibility is enabled to change the expected operating point, can contribute to resolve the network operational problems [3]. The power flexibility may be divided for active or reactive power, which can assist in the network problems at a certain cost. This will ensure that the DSO can maintain the ability of granting network access to consumers and producers, with power quality, safety, and stability.

The transmission system operator (TSO) and DSO coordination may be a path to explore as DSO finds here the opportunity to coordinate a reactive power service with the TSO. This coordination

intends to avoid voltage and/or congestion problems in the transmission system assuring that the distribution continues working without problems. Depending on the policies and agreements, this service can be remunerated given an extra advantage to the DERs. In fact, preventive reactive power management models are emerging as a potential solution for improving the coordination between the DSO and the TSO, while ensuring proper levels of voltage control in the system, as shown in [4].

There are many literature examples regarding approaches on reactive power management with fixed active power injection as in [5,6]. These, however, do not consider networks with strong RES penetration. A stochastic approach for ensuring voltage stability is proposed in [7]. The method considers a two-stage stochastic model with multi-objectives, such as minimization of power losses, operation, and management costs and wind power costs. Similarly, [8] proposes a stochastic model for corrective voltage control under severe contingencies, considering the uncertainty of wind power producer and consumer demand. A coordinated active and reactive optimization of an active distribution network considering energy storage systems and relaxed optimal power flow is proposed in [9]. It proposes a multi-objective function for minimizing power losses, operation costs and voltage deviation, however, the reactive power provision to assist in the TSO/DSO coordination is disregarded.

Most of these works disregard the full behavior of the distribution grid, introducing approximations and linearized versions of the full alternative current optimal power flow (AC-OPF). This can lead to sub-optimal solutions that may be infeasible. Thus, [10] proposes a voltage sensitivity analysis for adjusting the reactive power setpoint of DER in order to improve voltage stability and provide reactive power to upper levels of the network. Complementarily, [11] models an adaptive control of the reactive power setpoints of wind farms to assist the TSO/DSO coordination, minimizing the losses while ensuring proper level of reactive power provision. Still, none of these works can schedule in advance adequate reactive power setpoints for DER, considering the uncertain and variable behavior of RES.

In this scope, the main objective of this paper is to propose a stochastic reactive power management model to assist the DSO in the reactive power management ahead of the operating hour. The main contributions of this paper are threefold:

- To design a two-stage stochastic reactive power management model considering a full AC-OPF. It has the purpose of aiding the decision-making of the DSO under the uncertain and variable behavior of RES connected in the distribution network;
- To propose a reactive power service provided by the DSO to the TSO in advance of the operating hour. This service can be used by the TSO in the transmission system management, defining a reactive power operation in the TSO/DSO boundary substations. This can help the TSO in different services like the voltage control and congestion management in the transmission system;
- Take into account the Portuguese reactive power policy on distribution grids, assessing the behavior and applicability of the proposed model.

This paper is structured as follows: Section 2 describes the Portuguese reactive power policies and introduces the sequential AC-optimal power flow (SOPF) tool model; Section 3 presents the mathematical formulation of the stochastic approach for reactive power management; Section 4 validates the proposed model based on a 37-bus distribution network with real data; Section 5 presents the most important conclusions.

## 2. Reactive Power Policies

### 2.1. Portuguese Reactive Power Policy

The Portuguese reactive power policy for the distribution network is based on the total inductive and capacitive reactive power that a generating unit produces in an hour [12]. The reactive power is dependent on the active power injected by the generating unit in the form of $tan\,\phi$. The reactive power must have a deviation of less than $+/-5\%$ from the defined $tan\,\phi$.

Each day is divided into four periods: peak, full hours, valley and super valley. Yet, reactive power has only two classifications: peak and off-peak. There are two different schemes referring to the generating units. The ordinary scheme encompasses conventional units, whereas the special scheme, RES, industrial and urban waste, cogeneration and micro-producers. The ordinary scheme is limited to *tan* $\phi$ = 0.4 for the peak period and *tan* $\phi$ = 0 for off-peak hours.

For the special scheme, Table 1 illustrates the relationship between active and reactive power.

**Table 1.** Reactive power policy for the special scheme [12].

| Voltage Level | *tan* $\phi$ | |
|---|---|---|
| | Peak Period | Off-Peak Period |
| High Voltage | 0 | 0 |
| Medium Voltage (P > 6MW) | 0 | 0 |
| Medium Voltage (P ≤ 6MW) | 0.3 | 0 |
| Low Voltage | 0 | 0 |

Similarly, there is a contractual agreement between the TSO and the DSO whereby the upstream connection will have a profile of *tan* $\phi$ as regulated for the special scheme. This means that between seven–22 h the *tan* $\phi$ should be within −0.3 and 0.3 and in the remaining hours *tan* $\phi$ = 0, with a deviation of less than +/−5% of the *tan* $\phi$. Failure to do so results in a penalty applied to the entity responsible for the failure. This penalty is modelled in steps of *tan* $\phi$ infringement for peak hours. Equation (1) illustrates the current 3 steps violation of the limits penalties [13,14]:

$$0.3 \leq \tan \phi < 0.4,$$
$$0.4 \leq \tan \phi < 0.5, \tag{1}$$
$$0.5 \leq \tan \phi$$

with penalty factor applied to the reference price of reactive power of 0.33, 1 and 3 for each respective step.

### 2.2. Proactive Reactive Power Management

Though DERs are applying current reactive power policies, the distribution system does not take full advantage of DER's technical capabilities (especially RES) or consider their variable and intermittent behavior. More precisely, the uncertain DERs production is often overlooked, and their reactive power contribution is as well. This may lead to difficulties in reactive power management, especially in distribution networks with a high level of DER integration. In this scope, the SOPF tool proposed in this paper utilizes a two-stage stochastic reactive power management model (presented in Figure 1) to account for the uncertainty and variability related to the DER. The model has been developed for single period simulation using information from a representative set of scenarios for the DER. Stochastic optimization is used as a means to handle the uncertainty of DER. Such a probabilistic approach is integrated into the reactive power management problem through scenarios of active power generation of the DER. As DER's active power generation is uncertain, reactive power can be as well. Still, reactive power generation curve depends on the level of active power generation, and therefore, the reactive power can be constrained by *tan* $\phi$.

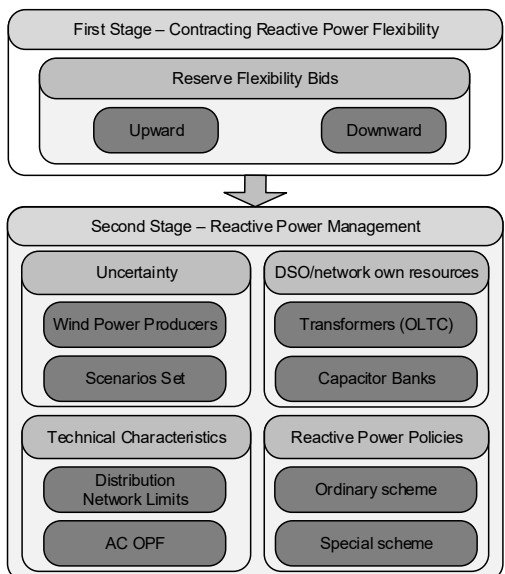

**Figure 1.** Two-stage optimization model embedded in the sequential AC-optimal power flow (SOPF) tool [15].

In the first stage, the DSO will contract in a day-ahead operating stage, the upward and downward reactive power flexibility that will be used during the operating stage. This represents the maximum reactive power fluctuation of the expected reactive operating point of the DER.

As for the operating stage, the second stage of the model, the DSO will activate the reactive power flexibility needed to overcome the system requirements, respecting the reactive power policy and the technical characteristics of the network (i.e., voltage and thermal limits).

The SOPF tool will then have as its output the amount and type of flexibility the DSO may need to operate the distribution system in expectation, ahead of the operating hour, outputting the upward and downward reactive power flexibility. The upward reactive power flexibility stands for increasing reactive power injection or decreasing reactive power absorbing in the grid, while downward reactive power flexibility stands for the opposite of the upward reactive power flexibility. Note that the reactive power flexibility should be contracted to the distributed generation to maintain the reactive power profile agreed with the TSO while outputting the optimal scheduling of the static equipment (capacitor banks and transformers with on-load tap changer—OLTC) managed by the DSO.

Nevertheless, the SOPF tool is scalable to consider distribution networks with hundreds of nodes, DER, and scenarios at the expense of computational effort. In addition, it does not consider network reconfiguration (which is a common tool for distribution grid management) but can be adapted to include it. Still, this would lead to increased complexity of the tool, hence the computational effort. Therefore, a tradeoff between system complexity and computational effort must be performed to ensure reliable solutions within the DSO operating window for day-ahead simulation.

## 3. Mathematical Formulation

The proactive reactive power management accounts for the uncertainty factor of renewable energy sources (RES). By utilizing the policies mentioned in Section 3.1, the proposed tool aims to manage the reactive power of the distribution network and to provide reactive power control according the TSO needs.

### 3.1. Objective Function

The objective Function (2) aims to minimize the operating costs of the DSO to maintain the distribution grid operating within the limits. It includes the costs related to each of the stages, in

which the first stage ($F^{DA}$) comprises the here-and-now decisions and the second-stage ($F^{RT}$) the wait-and-see decisions.

$$\min \quad F^{DA} + F^{RT} \tag{2}$$

where $F^{DA}$ and $F^{RT}$ are described as in (3) and (4).

$$F^{DA} = \sum_{g=1}^{N_G}\left(C_{DER(g)}^{Q,UP}R_{DER(g)}^{Q,UP} + C_{DER(g)}^{Q,DW}R_{DER(g)}^{Q,DW}\right) + p_{TSO}^{Q,UP}RLX_{TSO}^{Q,UP} + p_{TSO}^{Q,DW}RLX_{TSO}^{Q,DW} \tag{3}$$

$$F^{RT} = \sum_{\omega=1}^{\Omega}\pi_{(\omega)}\left[\begin{array}{l}\sum_{g=1}^{N_G}\left(C_{DER(g)}^{act}\left(r_{DER(g,\omega)}^{Q,UP} - r_{DER(g,\omega)}^{Q,DW}\right) + C_{DER(g)}^{cut}P_{DER(g,\omega)}^{cut}\right) + p_{TSO}^{rlx,act}\left(rlx_{TSO(\omega)}^{Q,UP} - rlx_{TSO(\omega)}^{Q,DW}\right) + p_{TSO(\omega)}^{Extra}rlx_{TSO(\omega)}^{Extra} + \\ \sum_{l=1}^{N_L}\left(C_{(l)}^{DR}P_{(l,\omega)}^{DR}\right) + \sum_{cb=1}^{N_{CB}}\sum_{lv=1}^{N_{levels}}C_{CB(cb)}Z_{CB(cb,\omega,lv)} + \sum_{trf=1}^{N_{TRF}}\sum_{lv=1}^{N_{levels}}C_{TRF(trf)}Z_{TRF(trf,\omega,lv)}\end{array}\right] \tag{4}$$

$F^{DA}$ represents the first-stage decision of contracting reactive power flexibility. Here, DER provides cost inflicted, upward and downward reactive power flexibility. It is mathematically presumed that the TSO request of reactive power may be needed to be relaxed (represented by *RLX*). This mathematical relaxation proposes the possibility of a certain deviation of the requested *tan ϕ* value in the upstream connection and is affected by its own penalty. The *tan ϕ* value is dependent on agreements between the TSO and the DSO.

Concerning $F^{RT}$, it portrays the real-time operating costs of the distribution network. By the cost of an activation price, generators may change their reactive power operating point. In cases of higher need of flexibility (when the DSO cannot entirely provide the service), a different relaxation is activated through the binary variable $rlx^{Extra}$ allowing the DSO to provide part of the TSO request. By applying an even greater cost, it is possible to curtail the generators active power for relaxing situations where active power is creating problems in the distribution network. Demand response can also be contemplated to decrease the active power consumption, which in turn will reduce the reactive power consumption, under even greater penalties for this relaxation. These alternatives options will ensure that DSO prioritizes DER and consumers over providing the reactive power service to the TSO.

Capacitor banks and the transformers OLTC ability are also considered with a cost related to the lifetime degradation of the equipment by changing the tap set point [16].

### 3.2. First-Stage Constraints

The first-stage constraints, seen in (5) and (6), represent the DER flexibility for upward and downward reactive power. Similar constraints are applied to the mathematical relaxation of the external supplier flexibility.

$$R_{DER(g)}^{Q,UP,Min} \le R_{DER(g)}^{Q,UP} \le R_{DER(g)}^{Q,UP,Max}, \quad \forall g \in \{1,\ldots,N_G\} \tag{5}$$

$$R_{DER(g)}^{Q,DW,Min} \le R_{DER(g)}^{Q,DW} \le R_{DER(g)}^{Q,DW,Max}, \quad \forall g \in \{1,\ldots,N_G\} \tag{6}$$

### 3.3. Second-Stage Constraints

The second-stage constraints refer to the operating stage constraints that are introduced by the uncertainty of RES production. DER active power relates to its operating point for the energy schedule. This value is assumed as fixed by the conditional mean forecast for active power generation. This leads to the active power curtailment in the operating stage to be limited by:

$$P_{DER(g,\omega)}^{cut} \le P_{DER(g)}^{op} + \Delta P_{DER(g,\omega)}, \quad \forall g \in \{1,\ldots,N_G\}, \forall \omega \in \{1,\ldots,\Omega\} \tag{7}$$

The difference of active power between the realization scenario and the expected forecast in each scenario is represented as $\Delta P$. The active power flowing from the upstream connection (TSO) is limited

by the contracted boundaries between the TSO and the DSO and by the capacity of the transformers at the substation interconnection. Active power can be injected/absorbed by the TSO as seen in (8).

$$-P_{TSO(\omega)}^{Max} \le P_{TSO(\omega)} \le P_{TSO(\omega)}^{Max}, \quad \forall \omega \in \{1, \dots, \Omega\} \tag{8}$$

In addition, the second-stage also includes the bounds of the second-stage variables and the non-anticaptivity constraints, given by:

$$r_{DER(g,\omega)}^{Q,UP} \le R_{DER(g)}^{Q,UP}, \quad \forall g \in \{1, \dots, N_G\}, \forall \omega \in \{1, \dots, \Omega\} \tag{9}$$

$$r_{DER(g,\omega)}^{Q,DW} \le R_{DER(g)}^{Q,DW}, \quad \forall g \in \{1, \dots, N_G\}, \forall \omega \in \{1, \dots, \Omega\} \tag{10}$$

Constraints (9) and (10) are also applied to the mathematical relaxation represented through external suppliers.

Each DER has the possibility to provide inductive or capacitive reactive power under the operation limits defined in the Portuguese regulation.

$$-\left(P_{DER(g)}^{op} + \Delta P_{DER(g,\omega)} - P_{DER(g,\omega)}^{cut}\right)\tan\phi \le Q_{DER(g)}^{op} + r_{DER(g,\omega)}^{Q,UP} - r_{DER(g,\omega)}^{Q,DW} \le \left(P_{DER(g)}^{op} + \Delta P_{DER(g,\omega)} - P_{DER(g,\omega)}^{cut}\right)\tan\phi,$$
$$\forall g \in \{1, \dots, N_G\}, \forall \omega \in \{1, \dots, \Omega\} \tag{11}$$

In (12) and (13), it is represented the upward/downward activation of the mathematical relaxation for the TSO. This relaxation considers a high penalty because the main goal is to provide the service for the TSO request.

$$rlx_{TSO(\omega)}^{Q,UP} \le RLX_{TSO}^{Q,UP}, \quad \forall \omega \in \{1, \dots, \Omega\} \tag{12}$$

$$rlx_{TSO(\omega)}^{Q,DW} \le RLX_{TSO}^{Q,DW}, \quad \forall \omega \in \{1, \dots, \Omega\} \tag{13}$$

As a last resource to find a solution for congestion and voltage problems, demand response is used by the DSO, being constrained by:

$$P_{L(l,\omega)}^{DR} \le P_{L(l)}, \quad \forall l \in \{1, \dots, N_L\}, \forall \omega \in \{1, \dots, \Omega\} \tag{14}$$

Then, the actual reactive power consumption of consumer *l* is given by:

$$Q_{L(l,\omega)} = \left(P_{L(l)} - P_{L(l,\omega)}^{DR}\right)\tan\phi, \quad \forall l \in \{1, \dots, N_L\}, \forall \omega \in \{1, \dots, \Omega\} \tag{15}$$

where *tan ϕ* can be settled at 0.3 as assumed in [17].

Regarding the capacitor banks and transformers with OLTC, these devices are owned by the DSO and located in the substation. This means that the DSO has the knowledge of their characteristics. Capacitor banks are used to provide reactive power being modelled by levels of reactive power as in (16) and (17).

$$Q_{CB(cb,\omega,lv)} = Q_{CB(cb,lv)}^{levels} X_{CB(cb,\omega,lv)}, \quad \forall cb \in \{1, \dots, N_{CB}\}, \quad \forall \omega \in \{1, \dots, \Omega\}, \forall lv \in \{1, \dots, N_{levels}\} \tag{16}$$

$$\sum_{lv=1}^{N_{levels}} X_{CB(cb,\omega,lv)} = 1, \quad \forall cb \in \{1, \dots, N_{CB}\}, \forall \omega \in \{1, \dots, \Omega\} \tag{17}$$

The cost of changing the tap of the capacitor banks is multiplied by $Z_{CB}$, which represents the difference between the tap selection in the present period with the previous one, which is constrained by:

$$X_{CB(cb,\omega,lv)}^{t-1} - X_{CB(cb,\omega,lv)} \le Z_{CB(cb,\omega,lv)}, \tag{18}$$

$$X_{CB(cb,\omega,lv)} - X_{CB(cb,\omega,lv)}^{t-1} \leq Z_{CB(cb,\omega,lv)}, \quad \forall cb \in \{1,\dots,N_{CB}\}, \forall \omega \in \{1,\dots,\Omega\}, \forall lv \in \{1,\dots,N_{levels}\} \quad (19)$$

The transformers with OLTC constraints for voltage control are modelled as:

$$\Delta V_{TRF(trf,\omega,lv)} = V_{TRF(trf,lv)}^{levels} X_{TRF(trf,\omega,lv)}, \quad \forall \omega \in \{1,\dots,\Omega\}, \forall trf \in \{1,\dots,N_{TRF}\}, \forall lv \in \{1,\dots,N_{levels}\} \quad (20)$$

$$\sum_{lv=1}^{N_{levels}} X_{TRF(trf,\omega,lv)} = 1, \quad \forall \omega \in \{1,\dots,\Omega\}, \forall trf \in \{1,\dots,N_{TRF}\} \quad (21)$$

$$V_{sb(\omega)} = V_{sb(\omega)}^{ref} + \sum_{lv=1}^{N_{levels}} \Delta V_{TRF(trf,\omega,lv)}, \quad \forall \omega \in \{1,\dots,\Omega\}, \forall trf \in \{1,\dots,N_{TRF}\} \quad (22)$$

where $\Delta V_{TRF}$ represents the voltage level to be activated in the transformer by the DSO. $V_{TRF}^{levels}$ is a parameter representative of all possible taps of the transformer, and $X_{TRF}$ is the binary variable for selection of a unique tap level. $V_{sb}^{ref}$ is the reference of voltage magnitude at the substation before the use of OLTC ability by the transformer, while the final voltage value at the substation is denoted by $V_{sb}$. In addition, the cost for changing the tap of the transformer is included in the objective function (5), where $Z_{TRF}$ is the linearization of the absolute function, as the capacitor banks. Thus, the constraints are:

$$X_{TRF(trf,\omega,lv)}^{t-1} - X_{TRF(trf,\omega,lv)} \leq Z_{TRF(trf,\omega,lv)}, \quad (23)$$

$$X_{TRF(trf,\omega,lv)} - X_{TRF(trf,\omega,lv)}^{t-1} \leq Z_{TRF(trf,\omega,lv)}, \forall trf \in \{1,\dots,N_{TRF}\}, \forall \omega \in \{1,\dots,\Omega\}, \forall lv \in \{1,\dots,N_{levels}\} \quad (24)$$

Moreover, an AC-OPF is used to model the power flow in the distribution network. Therefore, the active power balance in each bus is modelled as:

$$\sum_{g=1}^{N_G} \left( P_{DER(g)}^{op,i} + \Delta P_{DER(g,\omega)}^i - P_{DER(g,\omega)}^{cut} \right) + P_{TSO}^i + \sum_{l=1}^{N_L} \left( P_{L(l,\omega)}^{DR,i} - P_{L(l)}^i \right) = G_{ii} V_{i(\omega)}^2 + V_{i(\omega)} \sum_{j \in TL^i} V_{j(\omega)} \left( G_{ij} \cos \theta_{ij(\omega)} + B_{ij} \sin \theta_{ij(\omega)} \right),$$
$$\forall i \in \{1,\dots,N_{Bus}\}, \forall \omega \in \{1,\dots,\Omega\}, \theta_{ij(\omega)} = \theta_{i(\omega)} - \theta_{j(\omega)} \quad (25)$$

Additionally, the reactive power balance is given by:

$$\sum_{g=1}^{N_G} \left( Q_{DER(g,\omega)}^{op,i} + r_{DER(g,\omega)}^{Q,UP,i} - r_{DER(g,\omega)}^{Q,DW,i} \right) - \sum_{l=1}^{N_L} Q_{L(l,s)}^i + \sum_{cb=1}^{N_{CB}} \sum_{lv=1}^{N_{levels}} Q_{CB(cb,\omega,lv)}^i + Q_{TSO(\omega)}^{op,i} + rlx_{TSO(\omega)}^{Q,UP,i} - rlx_{TSO(\omega)}^{Q,DW,i} + rlx_{TSO(\omega)}^{Extra,i} =$$
$$V_{i(\omega)} \sum_{j \in TL^i} V_{j(\omega)} \left( G_{ij} \sin \theta_{ij(\omega)} - B_{ij} \cos \theta_{ij(\omega)} \right) - B_{ii} V_{i(\omega)}^2, \quad \forall i \in \{1,\dots,N_{Bus}\}, \forall \omega \in \{1,\dots,\Omega\}, \theta_{ij(\omega)} = \theta_{i(\omega)} - \theta_{j(\omega)} \quad (26)$$

There is also the consideration that the energy flowing through the distribution lines has a thermal limit that should not be exceeded, being limited as in (27) and (28).

$$\left| \overline{V_{i(\omega)}} \left[ \overline{y_{ij}} \overline{V_{ij(\omega)}} + \overline{y_{sh(i)}} \overline{V_{i(\omega)}} \right]^* \right| \leq S_{TL}^{Max}, \overline{V_{ij(\omega)}} = \overline{V_{i(\omega)}} - \overline{V_{j(\omega)}}, \quad \forall i,j \in \{1,\dots,N_{Bus}\}, \forall \omega \in \{1,\dots,\Omega\}, i \neq j \quad (27)$$

$$\left| \overline{V_{j(\omega)}} \left[ \overline{y_{ij}} \overline{V_{ji(\omega)}} + \overline{y_{sh(j)}} \overline{V_{j(\omega)}} \right]^* \right| \leq S_{TL}^{Max}, \overline{V_{ji(\omega)}} = \overline{V_{j(\omega)}} - \overline{V_{i(\omega)}}, \forall i,j \in \{1,\dots,N_{Bus}\}, \forall \omega \in \{1,\dots,\Omega\}, i \neq j \quad (28)$$

Voltage magnitude must stay between the limits established by the DSO, assuming the slack bus voltage magnitude as fixed.

$$V_{Min}^i \leq V_{i(\omega)} \leq V_{Max}^i, \quad \forall \omega \in \{1,\dots,\Omega\} \quad (29)$$

## 4. Case Study

This section presents the case study used to apply and test the model developed for the Portuguese reactive management policies. The simulation has been carried out with MATLAB and GAMS tools.

### 4.1. 37-Bus Distribution System

The present case study is based on a 37-bus distribution network (originally presented in [18]) that was adapted to support five DER in the form of three combined heat and power (CHPs) and two wind turbines, as one can see in Figure 2.

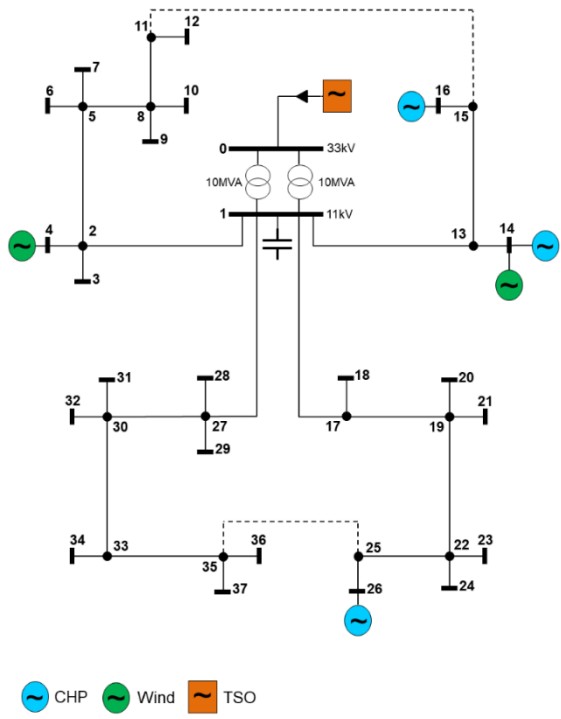

**Figure 2.** 37-Bus distribution network (adapted from [18]).

The distribution network is connected to a high voltage network through two power transformers of 10 MVA each. It also possesses 22 consumption points that represent 1908 consumers (1850 residential consumers, two industrial consumers, 50 commercial stores, and six service buildings) [18], with the consumption characteristics and profile being adopted from [19]. The total active and reactive power consumption in the network is summarized in Table 2.

There are two transformers and two capacitor banks being considered in the network. More precisely, the transformers have OLTC ability with a maximum voltage deviation of 0.1 p.u. In addition, the capacitor banks also have steps with a total capacity of reactive power production of 5.4 MVAr (one capacitor bank with 4.5MVAr and the other with 0.9 MVAr). The cost reflecting the use of the transformers and capacitor banks (with the OLTC ability that reduces the equipment lifetime) is determined by [16]. It is assumed that both types of equipment are owned and managed by the DSO.

The network is composed of different DER. More precisely, three CHP units and two wind turbines which can provide reactive power flexibility, accordingly to their technical limits.

It is assumed that the DER active power generation should be fully absorbed by the network, following the standard regulation. Therefore, Table 3 shows the generic characteristics of the DER, including the expected operating point (e.g., wind power forecast).

The DER characteristics, as well as the cost for upward and downward reactive power flexibility, are given in Table 4.

**Table 2.** Active and reactive power consumption characteristics.

| Load.Bus | | Active Power Consumption (kW) | | | Reactive Power Consumption (kVAr) | | |
|---|---|---|---|---|---|---|---|
| | | Min | Mean | Max | Min | Mean | Max |
| 1 | 3 | 373.2 | 677.9 | 1190.5 | 112.0 | 203.4 | 357.2 |
| 2 | 4 | 206.1 | 591.2 | 1015.6 | 61.8 | 177.4 | 304.7 |
| 3 | 6 | 88.4 | 599.0 | 1029.8 | 26.5 | 179.7 | 308.9 |
| 4 | 7 | 394.7 | 716.9 | 1259.1 | 118.4 | 215.1 | 377.7 |
| 5 | 9 | 539.0 | 761.8 | 1089.0 | 161.7 | 228.5 | 326.7 |
| 6 | 10 | 298.7 | 636.6 | 1040.9 | 89.6 | 191.0 | 312.3 |
| 7 | 12 | 323.0 | 586.5 | 1030.1 | 96.9 | 176.0 | 309.0 |
| 8 | 14 | 387.0 | 1110.4 | 1907.4 | 116.1 | 325.4 | 567.1 |
| 9 | 16 | 745.6 | 1589.1 | 2598.3 | 223.7 | 425.6 | 779.5 |
| 10 | 18 | 509.7 | 720.3 | 1029.8 | 152.9 | 169.7 | 308.9 |
| 11 | 20 | 88.4 | 599.0 | 1029.8 | 26.5 | 152.1 | 308.9 |
| 12 | 21 | 373.2 | 677.9 | 1190.5 | 112.0 | 190.9 | 357.2 |
| 13 | 23 | 365.1 | 778.1 | 1272.3 | 109.5 | 208.4 | 381.7 |
| 14 | 24 | 539.0 | 761.8 | 1089.0 | 161.7 | 179.5 | 326.7 |
| 15 | 26 | 323.0 | 586.5 | 1030.1 | 96.9 | 165.1 | 309.0 |
| 16 | 28 | 178.3 | 511.6 | 878.8 | 53.5 | 149.9 | 261.3 |
| 17 | 29 | 74.4 | 503.8 | 866.2 | 22.3 | 128.0 | 259.9 |
| 18 | 31 | 314.0 | 570.2 | 1001.4 | 94.2 | 160.5 | 300.4 |
| 19 | 32 | 290.4 | 618.9 | 1011.9 | 87.1 | 165.8 | 303.6 |
| 20 | 34 | 93.5 | 633.4 | 1089.0 | 28.1 | 160.9 | 326.7 |
| 21 | 36 | 217.9 | 625.3 | 1074.1 | 65.4 | 183.2 | 319.3 |
| 22 | 37 | 323.0 | 586.5 | 1030.1 | 96.9 | 165.1 | 309.0 |

**Table 3.** General characteristics and operating point for distributed energy resources (DER).

| DER | Number of Units | Total Installed Power | Operating Point $P^{op}$ (MW) | | |
|---|---|---|---|---|---|
| | | | Min | Mean | Max |
| CHP | 3 | 2.5 (MVA) | 1.0 | 1.15 | 1.5 |
| Wind | 2 | 20 (MVA) | 11.31 | 14.01 | 15.34 |
| Transmission system operator (TSO) | 1 | 20 (MVA) | - | - | - |

**Table 4.** DER reactive power costs.

| DER | Upward Cost $C^{up}$ (m.u./kVAr) | | | Downward Cost $C^{dw}$ (m.u./kVAr) | | |
|---|---|---|---|---|---|---|
| | Min | Mean | Max | Min | Mean | Max |
| CHP | 0.02 | 0.04 | 0.06 | 0.02 | 0.04 | 0.06 |
| Wind | 0.02 | 0.025 | 0.03 | 0.02 | 0.025 | 0.03 |
| TSO | 1 | 1 | 1 | 1 | 1 | 1 |

RESs are modelled through stochastic variables. Thus, upward and downward reactive power flexibility is constrained by their technical limits. In [20,21] can be found the scenarios used to model the uncertainty of wind power forecast. A set of 10 scenarios were extracted for each wind generator. In this case study, the standard reactive power policy of the DER is subjected to the Portuguese regulation, following Table 1.

Regarding the upstream connection, it must be established the *tan ϕ* agreed between the TSO and the DSO for the substation of interconnection. In this case, it has been considered that the *tan ϕ* varies throughout the day according to the regulation established in [12]. More precisely, the TSO must provide a *tan ϕ* of 0 with +/−5% of deviation between 22:00 and the 07:00. In the remaining period, the *tan ϕ* is expected to be 0.3 with +/−5% deviation. Note that the *tan ϕ* can vary from these values, taking into account specific agreement between the DSO and the TSO for a specific substation of interconnection. The case study was constructed assuming that active power from DER can be

greater or less than the load on the network. Thus, the TSO can either inject or absorb active power depending on the realization of wind generation over time.

*4.2. Results*

The tool will attempt, by using capacitor banks, transformers with OLTC and DER, to contract optimal reactive power flexibility and with this, meet the desired reactive power profile defined by the DSO.

It is important to note that active power production in the DERs is fixed according to the previous forecast of the energy dispatched for each hour. Wind power plants have a forecast point determined for the next 24 h along with 10 possible hourly scenarios, as in Figure 3.

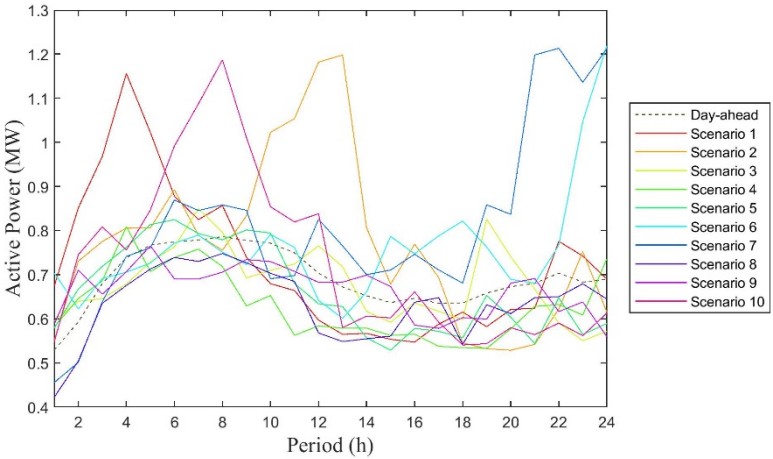

**Figure 3.** Point forecast and scenarios of the wind power plant 1 throughout 24 h.

Using wind active power as an input and considering the flexibility limits, the expected reactive power production operation is determined by the tool for each operation hour point. The *tan ϕ* of both wind power plants is kept within the +/−5% range in every scenario. Figure 4 shows the evolution of *tan ϕ* overtime for the wind power plant 1.

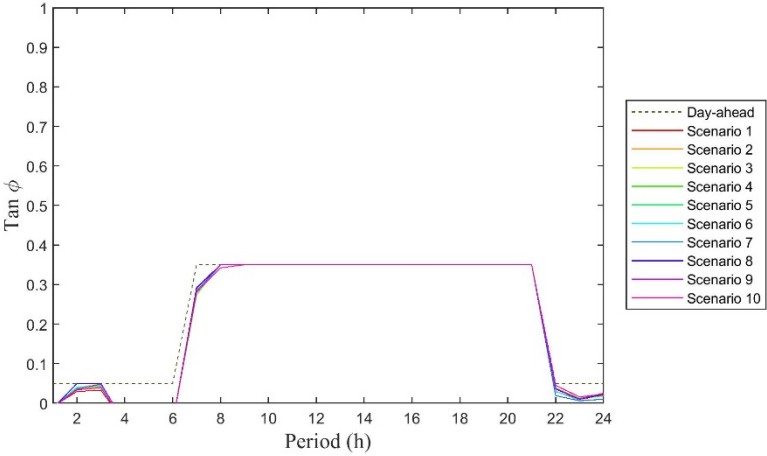

**Figure 4.** Forecasted and realized tan ϕ of the wind power plant 1 throughout 24 h.

Figure 5 depicts the tan ϕ profile at the substation. As was predicted, the *tan ϕ* values are as close as possible to 0.3 between 7–22 h and a value of 0 for *tan ϕ* for the other hours, for every scenario, to guaranty that no penalties to be applied to the DSO. DERs reactive power generation, the capacitor banks and the transformers with OLTC ability have an important role in securing this result.

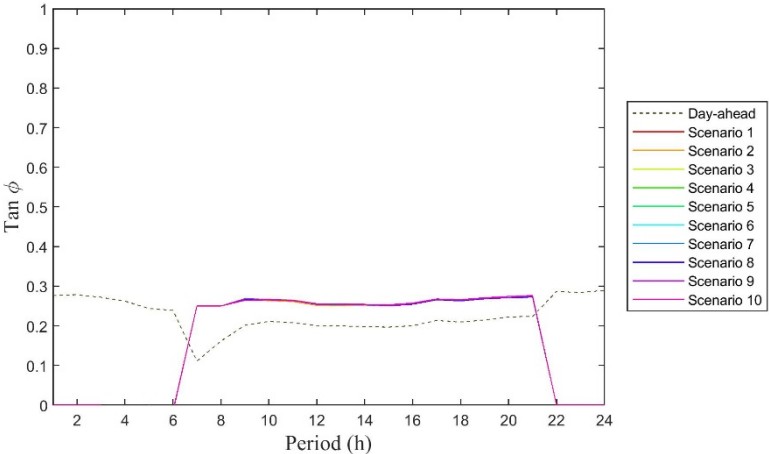

**Figure 5.** Forecasted and realized *tan* $\phi$ at the main substation throughout 24 h.

Figure 6 shows the reactive power production of capacitor bank 1. The level of reactive power varies according to the DSO needs. For periods between 22:00 and 07:00, the level of reactive power production is high, reaching the maximum production in some periods. During the day, the reactive power production of the capacitor bank comes to zero, since the TSO is injecting a significant amount of reactive power which is sufficient to support the system. It is also worth mentioning that the step position does not change more than four times per day, which reduces equipment degradation over time.

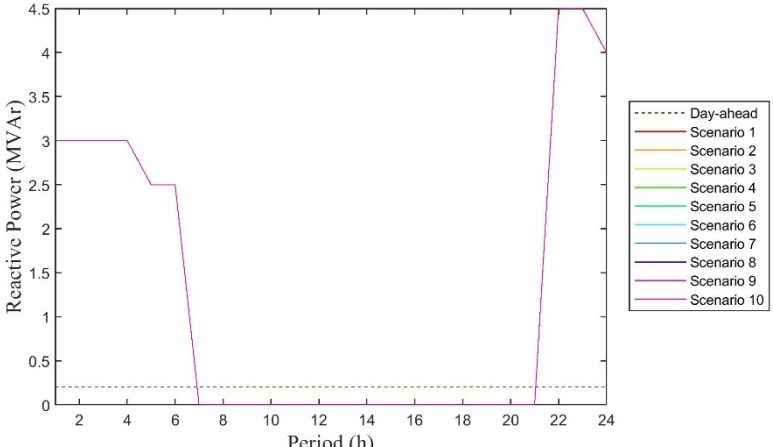

**Figure 6.** Reactive power production of capacitor bank 1 throughout 24 h.

Capacitor bank 2 follows the same behavior as capacitor bank 1, as can been seen in Figure 7. In fact, as capacitor bank 2 is much smaller than capacitor bank 1, the capacitor bank 2 is often used to complement the reactive power between steps of the capacitor bank 1.

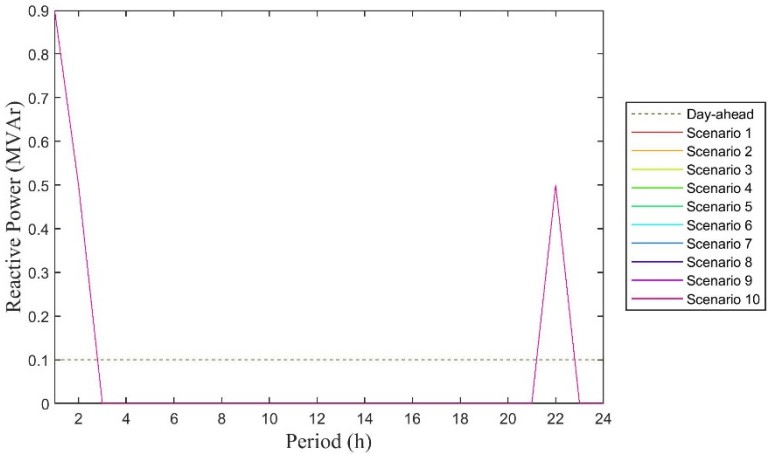

**Figure 7.** Reactive power production of capacitor bank 2 throughout 24 h.

As with the capacitor banks, OLTC tap changes are reduced, even maintaining the same position throughout the 24 h. It is noteworthy that as the optimization considers the day-ahead forecast point, it leads to the modification of the OLTC tap hourly positions (Figure 8). Note that both transformers present the same behavior as presented in Figure 8.

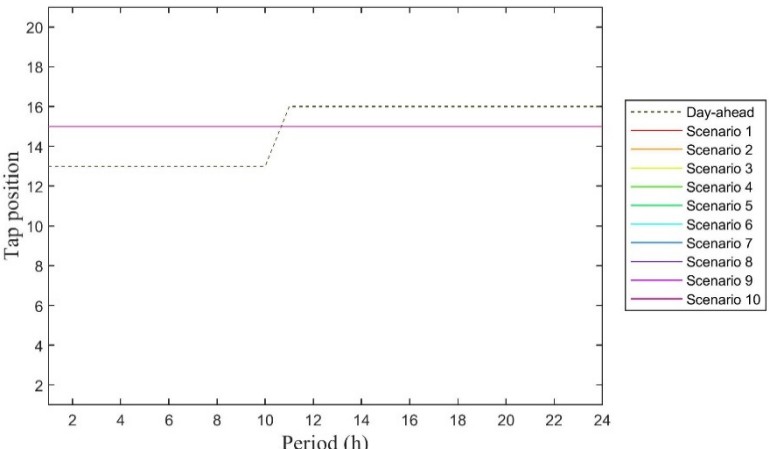

**Figure 8.** Transformer 1 with OLTC ability throughout 24 h.

## 5. Conclusions

This work proposes a new tool to be used by the DSO in reactive power management, exploiting DER flexibility. It explores the use of a two-stage stochastic model that manages the uncertainty of wind power producers. With this tool, TSO reactive power requirements can be provided by contracting the service to the DSO, which may be an alternative to investments in reactive power control equipment in the transmission network. Simulations were done for a 37-bus distribution network, whose results demonstrate the feasibility of the proposed tool. The selection of the DER that could provide reactive power flexibility, under the different operation conditions introduced, was proven and the service was provided to the TSO.

**Author Contributions:** Conceptualization, T.S. (Tiago Soares), L.C., H.M., T.S. (Tiago Simão) and M.L; Methodology, T.S. (Tiago Soares), L.C. and H.M.; Software, T.A. and T.S. (Tiago Soares); Validation, T.A., T.S. (Tiago Soares), L.C. H.M., T.S. (Tiago Simão) and M.L; Visualization, T.A. and T.S. (Tiago Soares); Writing—original draft, T.A., T.S. (Tiago Soares), L.C. and H.M., Writing—review & editing, T.A., T.S. (Tiago Soares), L.C., H.M, T.S. (Tiago Simão) and M.L.

**Funding:** This work was financed by the European Union's Horizon 2020 through the EU framework Program for Research and Innovation 2014–2020, within the EU-TDX-ASSIST project under the agreement No. 774500.

This work was also supported by national funds through Fundação para a Ciência e Tecnologia with reference UID/CEC/50021/2019.

**Conflicts of Interest:** The authors declare no conflict of interest. The funders had no role in the design of the study; in the collection, analyses, or interpretation of data; in the writing of the manuscript, or in the decision to publish the results.

## Nomenclature

**Parameters**

| | |
|---|---|
| $\Delta P$ | Power deviation in each scenario |
| $B$ | Imaginary part in admittance matrix |
| $C$ | Cost |
| $G$ | Real part in admittance matrix |
| $N$ | Number of unit resources |
| $p$ | Penalty for external supplier's flexibility |
| $\overline{y}$ | Series admittance of line that connects two buses |
| $\overline{y}_{sh}$ | Shunt admittance of line that connects two buses |

**Variables**

| | |
|---|---|
| $\theta$ | Voltage angle |
| $P$ | Active power |
| $Q$ | Reactive power |
| $r$ | Reactive power flexibility used in the operating stage |
| $rlx$ | Reactive power relaxation in the operating stage |
| $R$ | Reactive power flexibility contracted at day-ahead stage |
| $RLX$ | Reactive power relaxation at day-ahead stage |
| $S$ | Apparent power |
| $V$ | Voltage magnitude |
| $\overline{V}$ | Voltage in polar form |
| $V_{sb}$ | Voltage at slack bus |
| $\Delta V$ | Voltage level activated by the DSO in the transformer |
| $X$ | Binary variable |
| $Z$ | Auxiliary variable for absolute function linearization |

**Subscripts**

| | |
|---|---|
| $\omega$ | Index of scenarios |
| $cb$ | Index of capacitor bank units |
| $CB$ | Capacitor bank abbreviation |
| $g$ | Index of generators units |
| $i, j$ | Bus index |
| $l$ | Index of load consumers |
| $L$ | Load consumers abbreviation |
| $lv$ | Index of levels (tap changing) for capacitor banks and transformers |
| $TSO$ | Transmission system operator |
| $t$ | Time index |
| $trf$ | Index of transformer units |
| $TRF$ | Transformer abbreviation |

**Superscripts**

| | |
|---|---|
| $act$ | Activation cost of resources in real-time stage |
| $cut$ | Generation curtailment |
| $Max$ | Maximum limit |
| $Min$ | Minimum limit |
| $op$ | Operating point of the power resource |
| $Q, DW$ | Downward reactive power flexibility |
| $Q, UP$ | Upward reactive power flexibility |
| $DR$ | Demand response of consumer $l$ |

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
