# Peer review of "Reactive Power Management Considering Stochastic Optimization under the Portuguese Reactive Power Policy Applied to DER in Distribution Networks"

_energies, doi:10.3390/en12214028_

Round 1

Reviewer 1 Report

the main objective of the paper is not clearly stated. the significance contribution is not well defined.

English should be revised.

the focus of the paper and scope is not well aligned. I suggest changing title to be more close to actual scope of paper, including optimization.

the link between stochastic approach and power optimization is important, and author can discuss uncertainty quantification and how probabilistic approach can be integrated in the optimization. also in figure 1.

it is not clear the link between optimization and size or network configuration, more discussions are needed.

Reviewer 2 Report

The paper is very interesting and relevant to the current research in Reactive Power management in distribution systems . the method and theory is well presented as well as the motivation for the paper. It is important to show the management under real grid constraints and policies which is well shown here.

my only critic is the graphs in the simulation section in which all the colors of all senarios are to alike and therefore when there is places where the reader want to see the difference between the scenarios it is very difficult.

Reviewer 3 Report

Surprisingly, the authors use the term "reactive energy" that is nonsense. 

Could you explain how do you define this quantity and its physical meaning? Or could be that the authors incorrectly interchanged the terms of "power" and "energy"?

The problem formulation and derivation seem correct. The authors want to control tan (phi) using active power from DER's - and that is why reactive power plays a small role in the calculations.

Major problem seem to be the chosen 37-bus distribution system, In the description NO reactive power demand in the system is mentioned !. So is there a reactive load ? or Authors have forgotten to add it?

If such a load does not exist - so why did the authors choose a system that does not have reactive load - that is strange especially when the paper is about reactive power management.

Reactive power management  would make more sense also considering the reactive load (IEEE - 30 bus has such loads), this would also make the findings more valuable and more applicable to real life situations.

Round 2

Reviewer 1 Report

revised paper is fine

Reviewer 3 Report

The authors corrected the errors and sufficiently explained and modified the paper.

The paper shows now clear the aim, methods and proposed solution. After careful English correction , this article can be published.